# Investigation on cement-improved phyllite based on the vertical vibration compaction method

**Yingjun Jiang**, **Jiangtao Fan** *, **Yong Yi**[‡], **Tian Tian**[‡], **Kejia Yuan**, **Changqing Deng**

Key Laboratory for Special Area Highway Engineering of Ministry of Education, Chang' an University, Xi'an, China

☯ These authors contributed equally to this work.
‡ These authors also contributed equally to this work.
* 2020021070@chd.edu.cn

**Data Availability Statement:** All relevant data are within the paper and its Supporting information files.

**Funding:** This study was funded by the Transportation Technology plan project from Jilin

## Abstract

The vertical vibration compaction method (VVCM), heavy compaction method and static pressure method were used to form phyllite specimens with different degrees of weathering. The influence of cement content, compactness, and compaction method on the mechanical properties of phyllite was studied. The mechanical properties of phyllite was evaluated in terms of unconfined compressive strength ($R_c$) and modulus of resilience ($E_c$). Further, test roads were paved along an expressway in China to demonstrate the feasibility of the highly weathered phyllite improvement technology. Results show that unweathered phyllite can be used as subgrade filler. In spite of increasing compactness, phyllite with a higher degree of weathering cannot meet the requirements for subgrade filler. With increasing cement content, $R_c$ and $E_c$ of the improved phyllite increases linearly. $R_c$ and $E_c$ increase by at least 15% and 17%, respectively, for every 1% increase in cement content and by at least 10% and 6%, respectively, for every 1% increase in compactness. The higher the degree of weathering of phyllite, the greater the degree of improvement of its mechanical properties.

## 1. Introduction

The Qinba mountainous area of southern Shaanxi is rich in phyllite. But because phyllite is a kind of soft rock with low strength, poor stability and fragility [1–5], its unconfined compressive strength ($R_c$) and modulus of resilience ($E_c$) cannot meet the current subgrade design specifications. Therefore, it is not suitable for subgrade filling [6–9]. However, because of the lack of suitable fillers for subgrade filling along the line coupled with the inconvenience and difficulties in transportation in mountainous areas, projects in this region are compelled to use phyllite spoils for subgrade filling [10]. Local sourcing of materials and use of phyllite spoils as subgrade fillers will not only reduce the project cost and protect limited land resources, but also minimize the damage to the surrounding environment [11–14].

The feasibility of using phyllite as subgrade filler was explored in many domestic and international studies. Based on the research and analysis of the basic mineral composition, strength,

Provincial in the form of a grant (No. 2017ZDGC-7) awarded to YJ, the Scientific Research of Central Colleges of China for Chang'an University in the form of a grant (No. 300102218212) awarded to YJ, and the Scientific Project from Henan Provincial Communication in the form of a grant (No. 2020J-2-2 95) awarded to YJ.

**Competing interests:** The authors have declared that no competing interests exist.

and compaction characteristics of weathered soft rock, initially, the feasibility of using weathered soft rock to fill the subgrade was determined, and a preliminary determination method was proposed [15–19]. Yang et al. discussed the standard of soft rock as a filler and studied the influence of the weathering degree and rock structure on weathered soft rock through rock strength test and particle-breaking test [20]. In related work, the physical and mechanical properties of phyllite and determined the engineering characteristics such as the mineral composition and mechanical strength of phyllite [21–24]. Xiong and Xu conducted an experimental study on the water content and particle size distribution of weathered phyllite subgrade and concluded that phyllite-filled subgrade can meet the requirements for the stable settlement of the subgrade and the sharp drop in strength after the immersion of phyllite. Thus, the feasibility of using phyllite as one of the important bases for subgrade filling was established [25]. Experimental research on phyllite under different pressure levels showed that the pressure level of 600 kPa can meet the requirements of subgrade filling [26]. The improved filling technology of phyllite subgrade was studied in combination with the field test section, and it was found that after the road performance was greatly improved after phyllite was mixed with cement, thus fully meeting the requirements of subgrade filling, and hence, it can be regarded as qualified subgrade filling [6]. Zhu studied the gradation, strength, compaction characteristics, and CBR (California bearing ratio) of phyllite before and after improvement and concluded that phyllite can be used as subgrade filler after improvement [27]. Cen compared and analyzed the test situation of improving the phyllite subgrade with cements of different grades. It was discovered that the weathered phyllite improved by adding 3% cement can not only ensure the quality of highway subgrade engineering, but also save investment and reduce construction costs [1]. Vazquez used cement and lime to improve the phyllite and compared the strength characteristics and improvement effects of the two improved phyllite [28]. Mark added phyllite during the process of mixing cement concrete and studied the strength characteristics and growth law of concrete specimens [2]. Based on indoor classification, compaction, and bearing ratio tests, Gidigasu found that phyllite did not meet the specification requirements of subgrade fillers, but it could meet the requirements after improvement with 5% cement [29].

At present, the heavy compaction method is widely used for forming phyllite samples at home and abroad. But the aggregate is easy to be crushed when the specimen is formed by this method. Therefore, heavy compaction method is not suitable for aggregates with large particle size. In addition, when using the traditional heavy compaction method, is difficult to ensure the uniformity of the aggregate. With the continuous development of subgrade engineering machinery and equipment, especially the technology of road rollers, the heavy compaction method may no longer agree with the actual situation of the site. Therefore, in this study, the vertical vibration compaction method (VVCM) and heavy compaction method are compared to determine the method of indoor molding of specimens that is more in line with the actual field conditions so that the indoor test and the field compaction are more consistent. Moreover, the research on phyllite filling subgrade mainly focuses on the influence of cement content on the mechanical properties of the improved soil, and little research is reported on the influence of the compactness on the mechanical properties of the improved soil [30].

This paper takes China's Anping Expressway as the research object. The maximum dry density and optimal water content of weathered phyllite under different compaction methods are compared. The mechanical properties of the subgrade fillers of phyllite are studied from three aspects of cement content, compactness and compaction method. The construction technology for phyllite subgrade filling was determined. This technology has great practical significance and practical value.

**Table 1. Physical index of phyllite.**

| Phyllite type | Moisture content (%) | Particle density (g/cm$^3$) | Block density (g/cm$^3$) | Water absorption (%) | Porosity (%) |
|---|---|---|---|---|---|
| phyllite A | 1.9 | 2.726 | 2.702 | 1.01 | 2.83 |
| phyllite B | 3.3 | 2.701 | 2.678 | 2.62 | 3.96 |
| phyllite C | 4.5 | 2.682 | 2.657 | 4.59 | 8.57 |

## 2. Materials and methods

### 2.1 Phyllite sampling

The raw materials were taken from the phyllite found along the Anping expressway. According to the damage and hardness of the phyllite along the route, three representative types of phyllite, namely, A, B, and C were initially selected.

Phyllite A has a relatively complete rock mass, with a mainly plate-like structure, and the core is mostly a 10–15-cm-long block or column. The hammering sound is not clear, and the rock is brittle. The rock mass of phyllite B is relatively complete, mostly shaped as plates, blocks, or short columns. The core is mostly a 4–10-cm-long block with 3–4-cm-long fragments. The hammering sound is dull, and the rock is brittle. The parent rock structure of phyllite C has been destroyed, and the rock mass is strongly weathered and mostly scaly. After mechanical crushing, the core is mostly fragmenting of length 1–5 mm with a small number of fragments that are 1–2 cm long. The hammering sound is dull, and the rock is brittle and becomes muddy after being immersed in water.

The physical indexes of the A, B, and C types retrieved from the site were tested. The test results are listed in Table 1.

The data in Table 1 show that as the degree of weathering increases, the particle density and block density phyllite A, B, and C gradually decreased, while the water content, water absorption, and porosity gradually increase.

### 2.2 Cement

The cement selected is Shaanxi Jinlong brand P.O42.5, and the technical parameters of the cement are summarized in Table 2.

### 2.3 Test plans

**2.3.1 Specimen preparation.** The heavy compaction method currently used for compaction quality control does not match the rolling characteristics of on-site tools, and the heavy compaction method does not match the vibration compaction mechanism of the on-site roller. To address this problem, a VVCM that is more in line with the actual situation is introduced herein [31–34].

During the rolling operation of the vibratory roller, the eccentric block is driven by the vibrating shaft to rotate rapidly, forming the interference force of the vibration system of the "roller-compressed material.". Under the action of the interference force, the vibrating wheel

**Table 2. Technical parameters of the cement used in this study.**

| index | Specific surface area (m$^2$/kg) | Stability (mm) | 3d compressive strength (MPa) | 3d flexural strength (MPa) |
|---|---|---|---|---|
| Measured value | 331 | 1.0 | 23.1 | 5.8 |
| Standards | ≥300 | ≤5 | ≥17.0 | ≥3.5 |

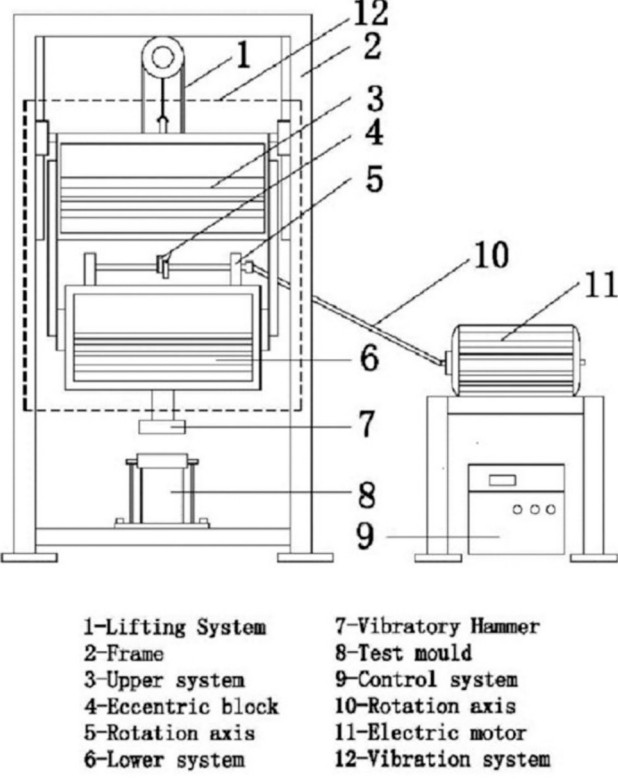

1-Lifting System
2-Frame
3-Upper system
4-Eccentric block
5-Rotation axis
6-Lower system

7-Vibratory Hammer
8-Test mould
9-Control system
10-Rotation axis
11-Electric motor
12-Vibration system

**Fig 1. Schematic of VVTE.**

vibrates with a frequency equal to that of the interference force [31–34]. The vibration is further transferred to the material to be pressed, making the material gradually denser.

In order to ensure the consistency of the indoor compaction effect and the on-site rolling effect and based on the working principle of the directional vibration roller, the vertical vibration testing equipment (VVTE) was used, as shown in Fig 1. Based on the existing research results [31–34], the parameters listed in Table 3 were used in the experiments.

The VVCM specimen was prepared as follows:

1. The phyllite filler is put into the oven, heated and dried at 105 ± 5°C for 4–6 h; subsequently, water was added for mixing.

2. The cushion block was placed in the lower part of the test mold, ensuring that the bottom was flat; the mixed phyllite sample was kept in the test mold, and the plunger was used for even insertion during the filling process.

3. The test mold with the phyllite sample was mounted in the VVTE, clamped using the control platform, and vibrated for 20s.

**Table 3. Vibration parameters.**

| Vibration frequency (Hz) | Vibration time (s) | Static pressure (kPa) | Weight (kg) | |
|---|---|---|---|---|
| | | | Upper system | Lower system |
| 40 | 20 | 40 | 108 | 167 |

4. After the compaction was completed, the height of the sample was measured to ensure that the height of the sample meets the specified requirements.

5. Finally, the sample was removed with a stripper.

The heavy compaction test and static pressure test were carried out in accordance with the relevant requirements in the Test Methods of Rock for Highway Engineering (JTG E41-2005) [35].

**2.3.2 Uniaxial compression test.** For uniaxial compressive strength testing, the rock sample must be a regular cylinder or cube. Since complete rock samples could be extracted only from weathered phyllite A, the uniaxial compressive strength test of this type of phyllite was performed in accordance with the Test Methods of Rock for Highway Engineering (JTG E41-2005) [35].

The regular saturated phyllite sample was placed at the center of the bearing plate of the pressure-testing machine to ensure that the sample, pressure plate, and ball seat surface were aligned with each other so that the sample received uniform force during pressure. When loading, the speed was maintained between 0.5 MPa/s and 1.0 MPa/s, and the test was stopped when the sample broke. The load at which the sample broke was recorded. During the test, a series of phenomena caused by the pressure failure of the sample was observed simultaneously.

The uniaxial compressive strength of the rock was determined according to Eq (1).

$$R = \frac{P}{A} \tag{1}$$

where $R$ is the uniaxial compressive strength of the specimen; $P$ is the maximum load of the damaged specimen; and $A$ is the cross-sectional area of the specimen.

A sample of weathered phyllite A was collected on site, and the uniaxial compressive strength test under water saturation was carried out to obtain the average value of uniaxial compressive strength. The results are listed in Table 4.

The data in Table 4 show that the uniaxial compressive strength value of weathered phyllite A after saturation was less than 15 MPa. According to Standard for engineering classification of rock mass (GB50218-2014) [36], this rock belongs to the category of soft rock.

**2.3.3 Point load test.** In the case of phyllite B and C, it was difficult to extract a complete rock sample on site owing to their high degree of weathering. Therefore, the uniaxial compression test cannot be used on these two phyllites, so only the point load strength test was used for these samples. Because the point load test has no specific requirements on the shape of the specimen, and it does not require that the specimen must be a regular cylinder or cube. The test is carried out with a point load tester, the rock sample is placed between the upper and lower load cones. Then, load is applied by means of the jack at uniform speed, and it is stopped when the sample breaks. Then, the point load strength is calculated.

The point load strength of the rock is determined according to Eq (2).

$$I_s = \frac{P}{D_e^2} \tag{2}$$

where $I_s$ is the uncorrected rock point load strength index; $P$ is the maximum load of the specimen when damaged; and $D_e$ is the equivalent core diameter.

**Table 4. Uniaxial compressive strength.**

| Phyllite type | Number of samples | Average uniaxial compressive strength (MPa) |
|:---:|:---:|:---:|
| Phyllite A | 10 | 8.73 |

**Table 5. Point load strength of phyllite.**

| Phyllite type | Point load strength test value (MPa) | | | | | Average value (MPa) |
|---|---|---|---|---|---|---|
| | **1** | **2** | **3** | **4** | **5** | |
| Phyllite A | 4.32 | 4.75 | 4.19 | 3.94 | 4.47 | 4.33 |
| Phyllite B | 2.29 | 2.17 | 2.06 | 2.35 | 2.11 | 2.20 |
| Phyllite C | 1.29 | 1.05 | 1.28 | 1.17 | 1.22 | 1.20 |

The point load strength test results of weathered phyllite A, B, and C samples in the saturated state are listed in Table 5.

The data in Table 5 show that in the saturated state, the point load strength of weathered phyllite samples of A, B, and C types decreases in that order. The main reason is that the porosity of phyllite increases with the degree of weathering, resulting in a decrease in the effective contact area that is reflected in the trend of decreasing strength.

**2.3.4 Modulus of resilience test.** According to the Test Methods of Soils for Highway Engineering (JTG E40-2007) [37], the test methods of $E_c$ include the bearing plate method and the strength meter method. In this study, the strength meter method was used for testing.

**2.3.5 *CBR* test.** The *CBR* test method adopted for the current specification [38] requires the specimen to be immersed in water for 96h. For weathered phyllite, it is difficult to meet the specification requirements. Considering only the requirements of the *CBR* will cause wastage of resources and serious ecological damage. In addition, the starting point of the standard *CBR* test method is to simulate the most unfavorable conditions during the use of the on-site subgrade materials, but often the standard test method does not match the actual situation of the subgrade site. In this study, using the existing research results, the *CBR* test method was improved with regard to three aspects: the soaking method, soaking time, and overlying pressure.

1. Soaking method: The upper soaking method is changed to side soaking. During the soaking process, the water level should be as high as possible, but the test tube should not be submerged.

2. Soaking time: Considering the permeability of phyllite, the soaking time is changed to 2 days.

3. Overburden pressure: When used to fill subgrade, upper subgrade, and lower subgrade, the overburden pressures used are 10, 20, and 30 kPa, respectively.

**2.3.6 Unconfined compressive strength test.** The $R_c$ of the specimen was tested according to the Specification for Design of Highway Subgrades (JTG D30-2015) [38].

## 3. Results and discussion

### 3.1 Comparison of VVCM and heavy compaction method

The weathered phyllite was subjected to heavy compaction and VVCM tests. The results are shown in Fig 2.

**3.1.1 Maximum dry density.** The comparison results of the maximum dry density of the on-site phyllite filler with the maximum dry density in VVCM and heavy compaction tests are listed in Table 6.

The data in Table 6 show that the maximum dry density on site is significantly higher than that obtained by VVCM, the compactness exceeds 100%. The main reason is that there are

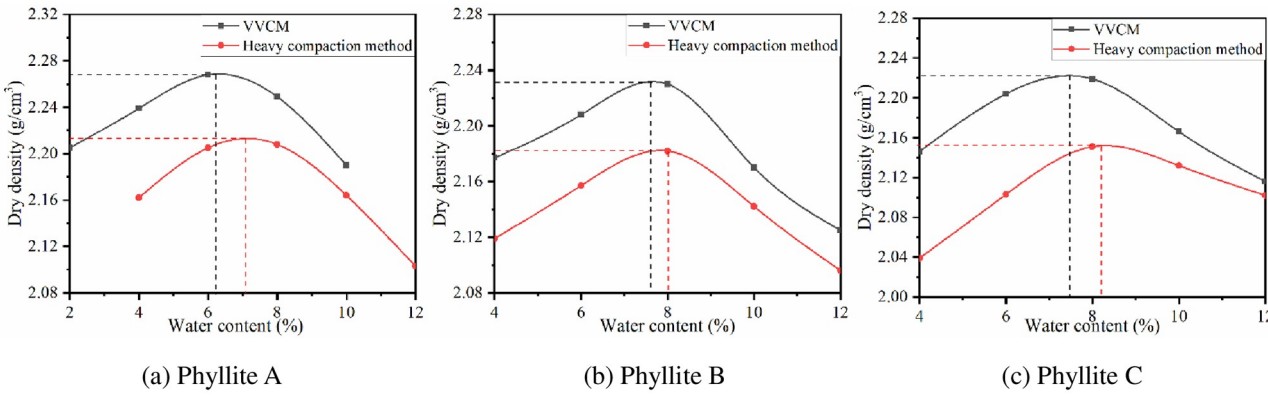

(a) Phyllite A  (b) Phyllite B  (c) Phyllite C

**Fig 2. Comparison of maximum dry density and optimal water content between heavy compaction and VVCM tests.**

**Table 6. Comparison of maximum dry density between on-site and indoor compaction tests.**

| Phyllite type | The maximum dry density (g/cm$^3$) | | | | |
|---|---|---|---|---|---|
| | On-site compaction | VVCM | Heavy compaction method | VVCM/ On-site compaction | VVCM/ Heavy compaction method |
| Phyllite A | 2.260 | 2.268 | 2.210 | 1.004 | 0.978 |
| Phyllite B | 2.221 | 2.231 | 2.182 | 1.005 | 0.982 |
| Phyllite C | 2.198 | 2.219 | 2.151 | 1.010 | 0.979 |

limitations to the compaction power and compaction method of heavy compaction, and hence, the phyllite filler fails to achieve high compactness. It also shows that the heavy compaction method is not consistent with the on-site vibration rolling method. In contrast, the VVCM gradually fills the pores between the large particles with fine particles through vibration, thereby improving the compactness of the sample. This is consistent with the characteristics of on-site vibration rolling, and the maximum dry density obtained based on VVCM has a high correlation with the field. Therefore, the VVCM is more compatible with on-site vibration rolling characteristics [32, 33].

**3.1.2 Optimal water content.** Specimens were formed by heavy compaction method and VVCM to determine the optimal water content. The results are listed in Table 7.

As seen from the data in Table 7, the optimal water contents of weathered phyllite A, B, and C determined by heavy compaction method are all higher than those determined by VVCM. This is because water mainly plays a role of lubrication during the compaction of subgrade fillers to reduce the resistance generated by the mutual compaction of particles. VVCM mainly uses vibration to rearrange and compact the phyllite particles, while heavy compaction method uses shear stress on the phyllite to further compact it. The frictional resistance that needs to be overcome in the heavy compaction process is greater, and the dependence on water lubrication

**Table 7. Comparison of optimal water content between heavy compaction method and VVCM.**

| Phyllite type | Optimal water content (%) | | |
|---|---|---|---|
| | Heavy compaction method | VVCM | Heavy compaction method/ VVCM |
| Phyllite A | 7.2 | 5.8 | 1.24 |
| Phyllite B | 8.2 | 7.0 | 1.17 |
| Phyllite C | 8.3 | 7.7 | 1.08 |

is higher, resulting in a larger optimal water content. Therefore, the best water content determined by VVCM should be used to control the construction on site.

## 3.2 Mechanical properties of phyllite subgrade filling

**3.2.1 Modulus of resilience.** *3.2.1.1 Influence of compaction method on $E_c$*. Under the optimal water content conditions, heavy compaction method and VVCM were used to form phyllite specimens with different compactness, and then the $E_c$ values of the specimens was tested in accordance with the Test Methods of Rock for Highway Engineering (JTG E41-2005) [35]. The results are summarized in Table 8. The compaction values listed in the table were calculated based on the maximum dry density of heavy compaction method.

The data in Table 8 show that as the compactness increases, $E_c$ of the phyllite subgrade filling also increases. The $E_c$ obtained by VVCM is generally greater than the heavy compaction method. For every 1% increase in compactness, $E_c$ of VVCM and heavy compacted phyllite A, B, and C samples increased by10%, 11%, and 13% on average. As the compactness increased, $E_c$ values obtained by the two compaction methods decreased continuously.

This behavior is attributed to the fact that as the compactness increases, the internal porosity of the specimen decreases, thereby increasing the effective area of the load. Owing to the differences in the mechanisms of the two compaction methods, the changes caused to the phyllite packing structure are greater in the case of VVCM. Under the action of vibration, the phyllite particles are more evenly distributed, and the pores in the sample are smaller. The friction and compaction between phyllite particles are correspondingly larger, and hence, $E_c$ of the phyllite formed by VVCM is larger than that of the phyllite formed by the heavy compaction method.

*3.2.1.2 Influence of compactness on modulus of resilience*. Under the optimal water content conditions, VVCM was used to form phyllite samples with different compactness. The results are shown in Fig 3.

Fig 3 shows that when the water content is constant, $E_c$ of phyllite increases linearly with compactness. When the specimen is formed by VVCM, for every 1% increase in compactness, $E_c$ of the weathered phyllite A, B, and C samples increased by 4.6, 3.2, and 2.1 MPa respectively; that is, for every 1% increase in compactness, $E_c$ increased by 15% to 17%. Thus, increasing the compactness can increase the $E_c$ correspondingly. The main reason is that while the compactness is increased, the porosity of the specimen is reduced, and the mechanical strength is also improved.

**3.2.2 CBR.** *3.2.2.1 Influence of compaction method on CBR*. Under the optimal water content conditions, heavy compaction method and VVCM were used to form phyllite specimens

**Table 8.** $E_c$ values of phyllite samples under different compaction methods.

| Compaction method | Phyllite A | | Phyllite B | | Phyllite C | |
|---|---|---|---|---|---|---|
| | Compactness (%) | $E_c$ value (MPa) | Compactness (%) | $E_c$ value (MPa) | Compactness (%) | $E_c$ value (MPa) |
| VVCM | 98.4 | 65.1 | 98.8 | 50.7 | 98.1 | 29.6 |
| | 99.8 | 72.5 | 99.6 | 53.5 | 99.8 | 34.5 |
| | 100.8 | 77.9 | 100.5 | 57.3 | 100.7 | 38.3 |
| | 102.5 | 90.6 | 102.7 | 66.4 | 102.9 | 43.2 |
| Heavy compaction method | 92.8 | 35.8 | 93.8 | 29.3 | 94.3 | 20.9 |
| | 96.7 | 42.6 | 96.5 | 35.8 | 96.1 | 22.1 |
| | 98.4 | 52.1 | 98.5 | 42.3 | 97.5 | 23.7 |
| | 99.8 | 67.0 | 99.8 | 50.2 | 99.7 | 31.5 |

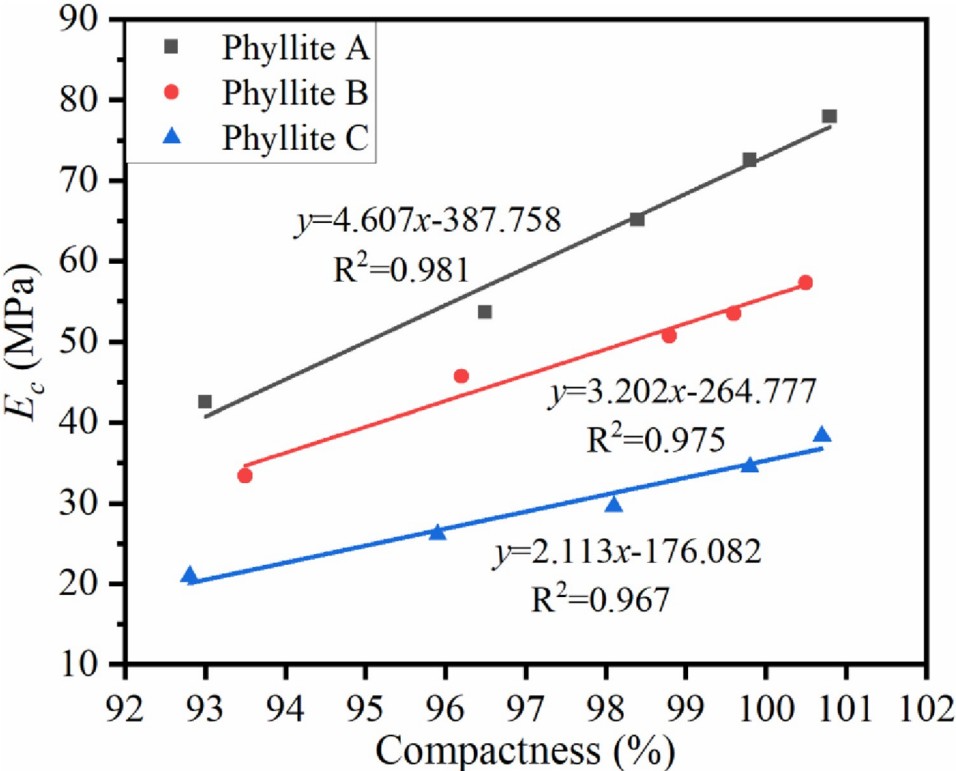

**Fig 3. The relationship between $E_c$ and compactness of the weathered phyllite.**

with different compactness. After the specimens were formed, the *CBR* test was performed according to the improved *CBR* test method. The test results are shown in Fig 4.

Fig 4 shows that as the compactness increases, the *CBR* of the phyllite subgrade filling too increases, and the *CBR* obtained by VVCM is greater than by the heavy compaction method. For every 1% increase in compactness, the *CBR* values of VVCM and heavy compaction phyllite increased by an average of 24% and 13%, respectively. Further, as the compactness increased, the ratio between the CBR obtained by the two molding methods too increased.

The main reason for this phenomenon is the differences between the mechanisms of the two compaction methods. In the heavy compaction method, the phyllite subgrade filler needs to overcome its own shear stress, which will result in more pores in the specimen. VVCM changes the internal structure of the phyllite filler, making the phyllite particles more evenly distributed. The pores in the specimen become smaller, forming a compact overall structure, and the friction and intercalation between the phyllite particles are correspondingly larger. Therefore, the strength of the phyllite formed by VVCM is greater than that of the sample formed by the heavy compaction method [34].

*3.2.2.2 Influence of compactness on CBR.* Under the optimal water content conditions, the heavy compaction method was used to form phyllite specimens with different compactness. The test was carried out according to the improved CBR method. The results are shown in Fig 5.

From Fig 5, it is clear that the *CBR* of the three weathered phyllite types all increase with the increase in compactness.

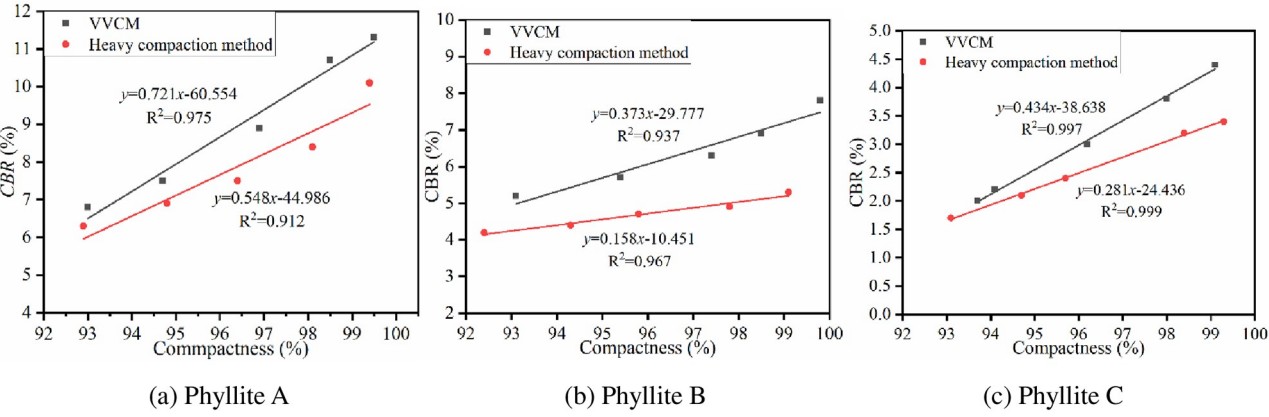

**Fig 4. *CBR* of the weathered phyllite samples under different compaction methods.**

1. Phyllite A can meet the requirements of subgrade filling under 2.7 kPa overburden pressure. Hence, only the relationship between compactness and *CBR* under 10 kPa overburden pressure was analyzed. Under the compactness required by specification, the *CBR* of phyllite A cannot meet the requirements, but it can meet the requirements when the compactness is increased to more than 97%.

2. The *CBR* of phyllite B meets the requirements for subgrade filling under the overburden pressure of 20 kPa and 30 kPa. When the compactness is increased to 98%, phyllite B can meet the *CBR* requirements of the lower subgrade at an overburden pressure of 10 kPa. When phyllite B is used as the upper subgrade filler, the compactness must reach 113% or more. It can be seen that when phyllite B is used as the upper subgrade filler, the compactness capacity of compaction equipment has been exceeded.

3. Under the compactness required by the specification, when phyllite C is used as the lower subgrade filler, its *CBR* value cannot meet the requirements. It can meet the requirements when the compactness reaches 97%. When phyllite C is used as the upper subgrade filler, the compactness needs to be increased to more than 100%. Hence, the dynamic compaction method must be used when phyllite C is used as the upper subgrade filler.

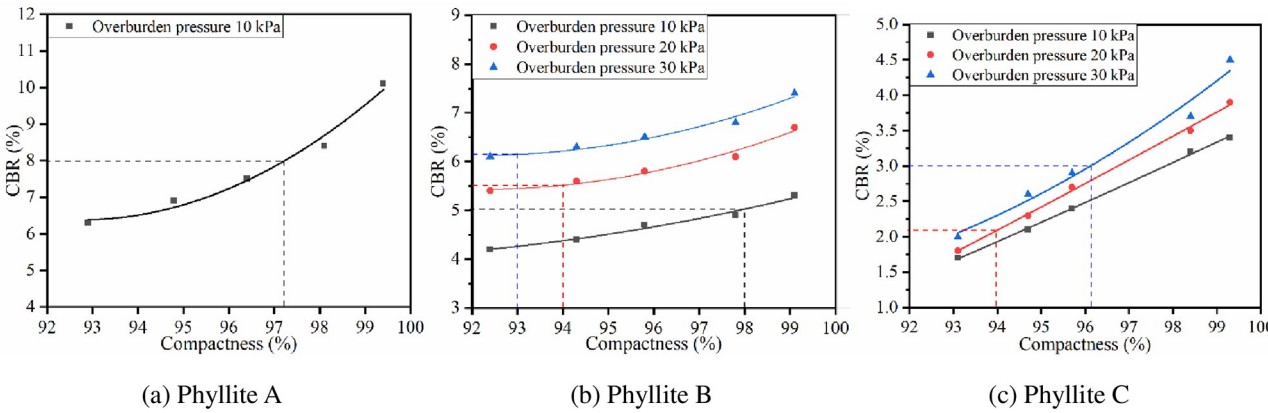

**Fig 5. Relationship between *CBR* and compactness of phyllites.**

## 3.3 Mechanical properties of cement-improved phyllite subgrade filling

From the *CBR* test results of phyllite subgrade filling, it is seen that even if the compactness is increased, the *CBR* of phyllite B still cannot meet the requirements of the upper subgrade, and the *CBR* of phyllite C cannot meet the requirements of upper subgrade and subgrade. Therefore, it is necessary to add an external admixture to improve the phyllite subgrade filler. In this test, cement was selected as the external admixture.

**3.3.1 Compaction characteristics of cement-improved phyllite.** According to Test Methods of Materials Stabilized with Inorganic Binders for Highway Engineering (JTG E51-2009) [39], VVCM and heavy compaction method were applied for three types of cement-improved phyllite, with the initial cement contents of 2%, 3%, 4%, and 5%. The relationship among the optimal water content, maximum dry density, and cement content of cement-improved phyllite is shown in Fig 6.

From Fig 6, it is seen that with an increase in cement content, the maximum dry density of the improved phyllite increased continuously. However, the increase rate was not obvious, indicating that the cement content has little effect on the maximum dry density of phyllite subgrade filler, but that it is still greater than the maximum dry density of unimproved phyllite subgrade filler. The optimum water content also shows an increasing trend with an increase in cement content. The main reason for this result is that in addition to the lubricating effect of water, the greater the amount of cement, the more is the amount of water required for cement hydration. In addition, the influence of compaction method on the maximum dry density and optimal water content of the improved phyllite is consistent with Section 3.1, further showing the advantages of VVCM.

**3.3.2 Mechanical properties of cement-improved phyllite.** *3.3.2.1 Unconfined compressive strength.*

(1) Cement content

The specimens were molded by VVCM under the optimal water content conditions, and tests were performed for cement contents of 2%, 3%, 4%, and 5%. After curing for 7 days under standard humidity and temperature conditions, $R_c$ was tested. The influence of cement content on $R_c$ of the improved phyllite subgrade filler is shown in Fig 7.

It can be seen from Fig 7 that the 7d $R_c$ of the improved phyllite shows an increasing trend with the increase in cement content. For every 1% increase in the cement content, $R_c$ increased by at least 15%. The largest increase in $R_c$ was observed for cement-improved phyllite A; for every 1% increase in cement content, its $R_c$ increases by 0.36 MPa. This is because with the increase of the cement content, the amount of new cementitious substances generated after the hydration reaction of the cement will also increase, effectively enhancing the bonding between the phyllite particles. The increase in cement content will cause more cement stones to be formed in the pores of the specimen, reduce the pores inside the specimen and increase its strength.

(2) Compactness

The relationship between the $R_c$ and the compactness of the improved phyllite is shown in Fig 8.

It can be seen from Fig 8 that when the cement content is constant, with an increase in compactness, $R_c$ of the improved phyllite subgrade filling shows a linear growth trend. For every 1% increase in compactness, $R_c$ of the improved phyllite increases by at least 11%. Phyllite B has the largest increase in $R_c$ when the cement content is 5%, and the $R_c$ increases by 0.12 MPa for every 1% increase in compactness. It can be seen that the compactness has a

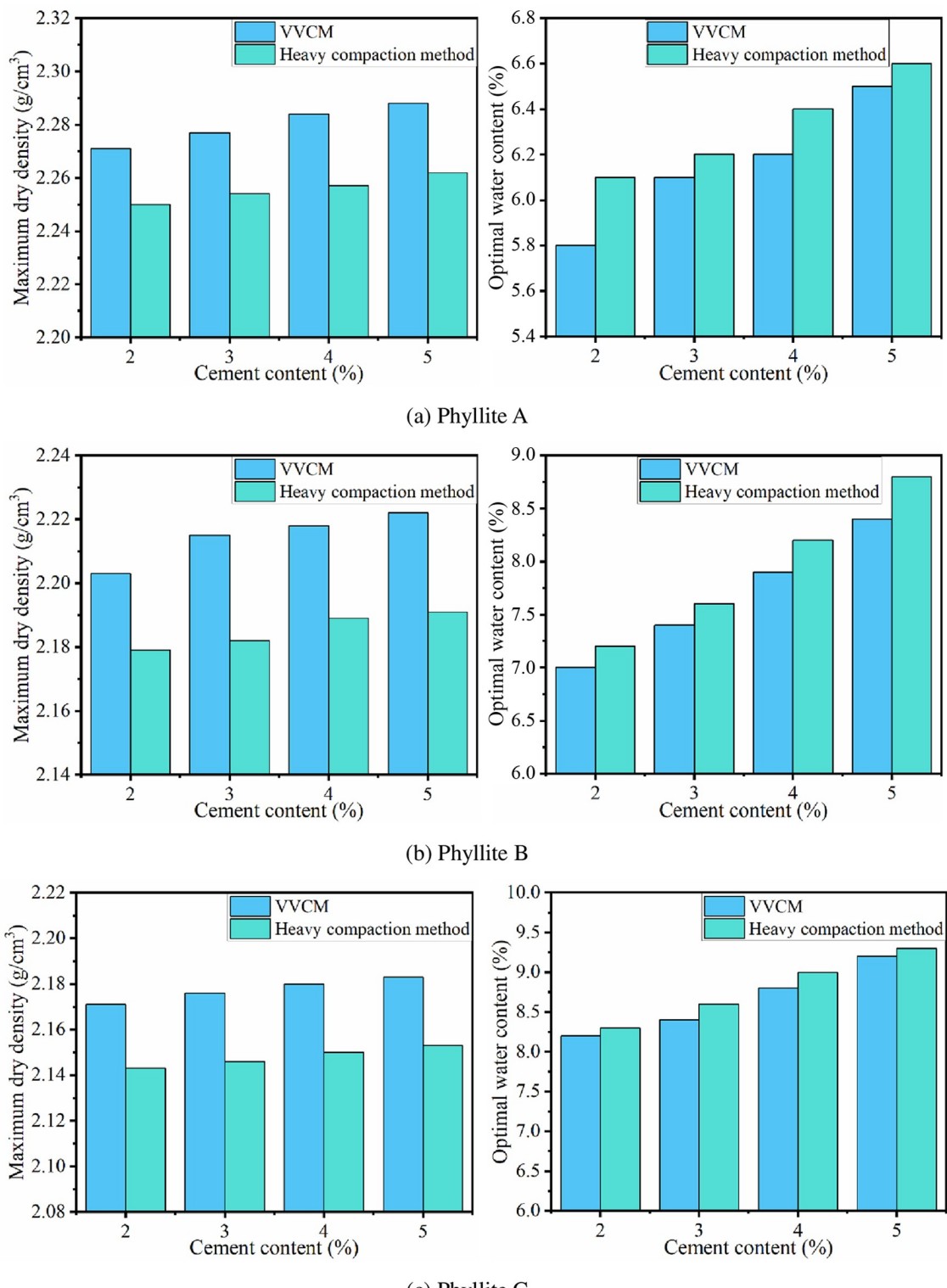

(a) Phyllite A

(b) Phyllite B

(c) Phyllite C

**Fig 6. Comparison of VVCM and heavy compaction method.**

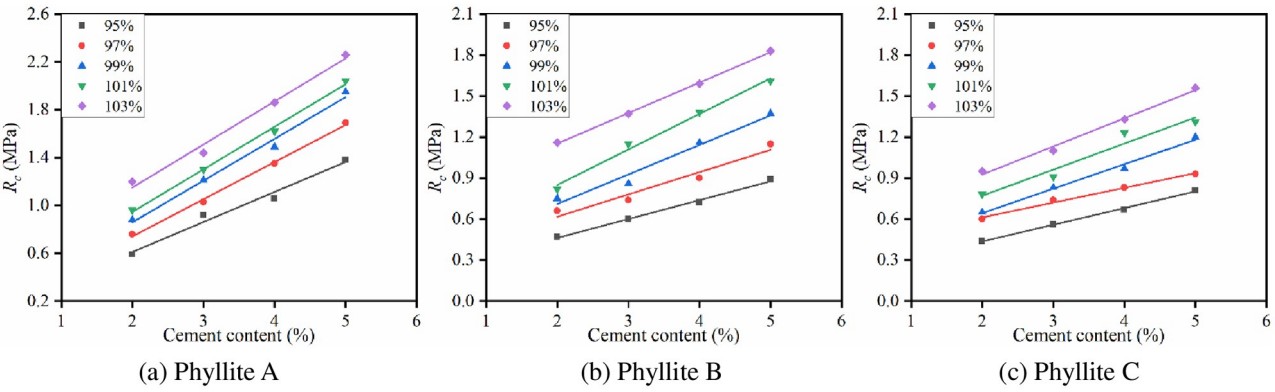

(a) Phyllite A　　　　　　　　(b) Phyllite B　　　　　　　　(c) Phyllite C

**Fig 7. Influence of cement content on $R_c$.**

significant effect on the $R_c$ of the improved phyllite subgrade filler. This mainly because of the increase in compactness, the porosity inside the specimen decreases, and the contact area of the specimen increases.

(3) Compaction method

VVCM and static pressure method were used to prepare cement-improved phyllite specimens with different compactness. The cement content was 4%, and the specimen was cured for 7 days under standard curing conditions, and the $R_c$ of the phyllite specimen was tested. The results are shown in Fig 9.

It can be seen from Fig 9, when the cement content was 4%, regardless of the compaction method, 7d $R_c$ shows a linear growth trend with the increase of compactness. Further, $R_c$ of improved phyllite specimens by VVCM can be increased by 15% to 23% for every 1% increase in compactness, which is far greater than the 12–18% of the static pressure method. It shows that the compaction method also has a certain influence on the $R_c$ of the improved phyllite subgrade filler. Phyllite particles are in a relatively flowing state under the action of vibration, and the resistance between the particles is reduced, thus forming a relatively dense state. The static pressure method simply relies on the vertical pressure to overcome the shear force between the particles, and it cannot make the particles fill each other to form a compact whole.

*3.3.2.2 Modulus of resilience.*

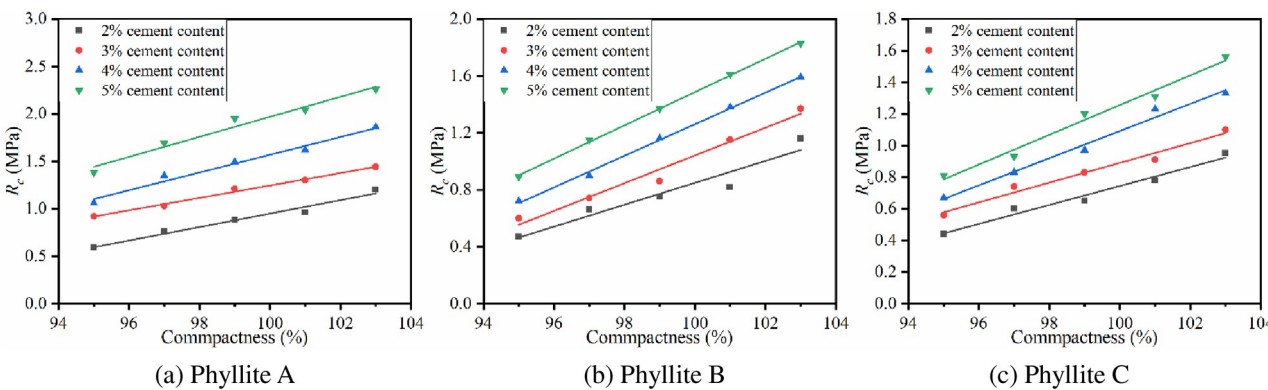

(a) Phyllite A　　　　　　　　(b) Phyllite B　　　　　　　　(c) Phyllite C

**Fig 8. The relationship between 7d $R_c$ and compactness.**

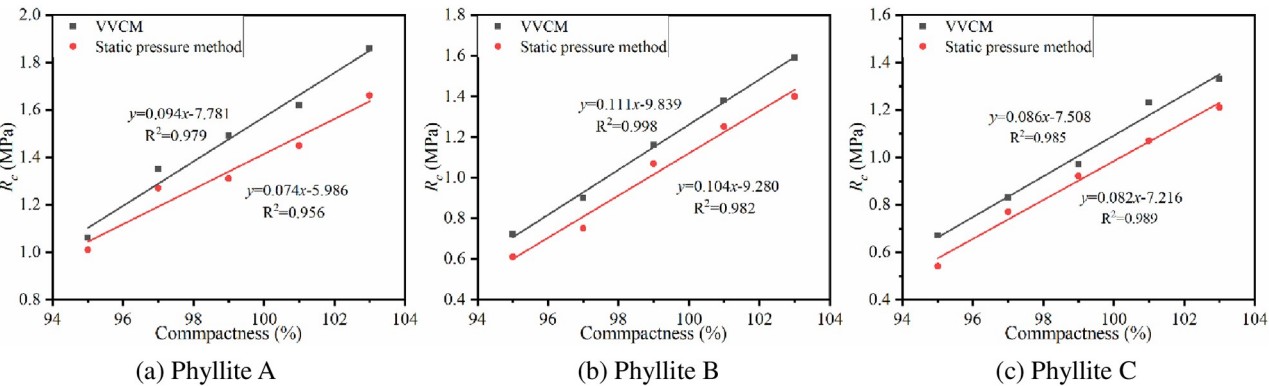

Fig 9. Relationship between 7d $R_c$ and compaction method.

(1) Cement content

The specimens were compacted by VVCM under the optimal water content conditions, and the cement contents were 2%, 3%, 4%, and 5%. After curing for 7 days under standard humidity and temperature conditions, the $E_c$ were determined. The influence of cement content on $E_c$ of improved phyllite subgrade filler is shown in Fig 10.

The figure shows that the 7d $E_c$ of the improved phyllite tends to increase with an increase in cement content. For every 1% increase in cement, its $E_c$ increases by at least 17%. The $E_c$ of cement-improved phyllite A has the largest increase. For every 1% increase in cement content, the $E_c$ increases by 62 MPa. This relationship is consistent with the $R_c$.

(2) Compactness

The relationship between 7d $E_c$ and compactness is shown in Fig 11.

The figure shows that when the cement content is constant, with an increase of compactness, the $E_c$ of the improved phyllite subgrade filling shows a linear growth trend. For every 1% increase in compactness, the $E_c$ of the improved phyllite increased by at least 6%. Phyllite B has the largest increase in $E_c$ when the cement content is 5%, and the $E_c$ increases by 19 MPa for every 1% increase in compactness. It can be seen that the compactness also has a significant effect on the $E_c$ of the improved phyllite subgrade filler.

(3) Compaction method

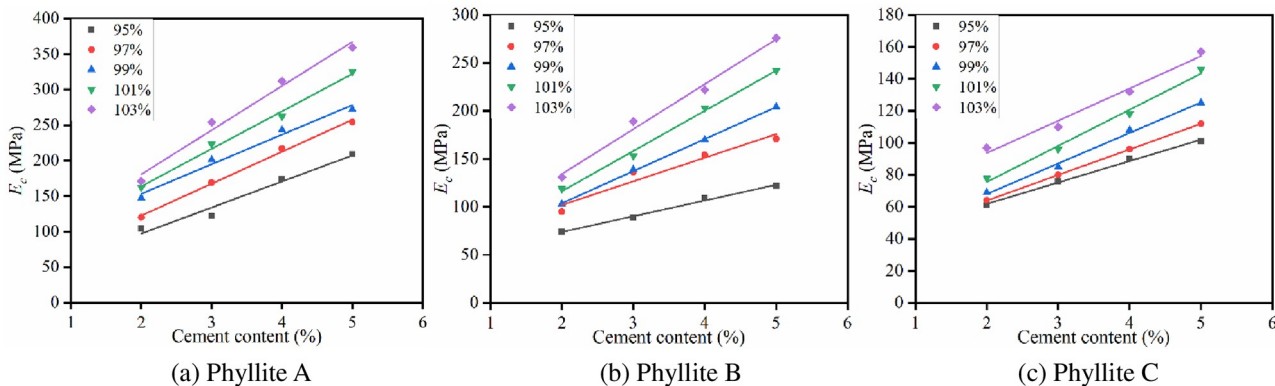

Fig 10. The influence of cement content on $E_c$.

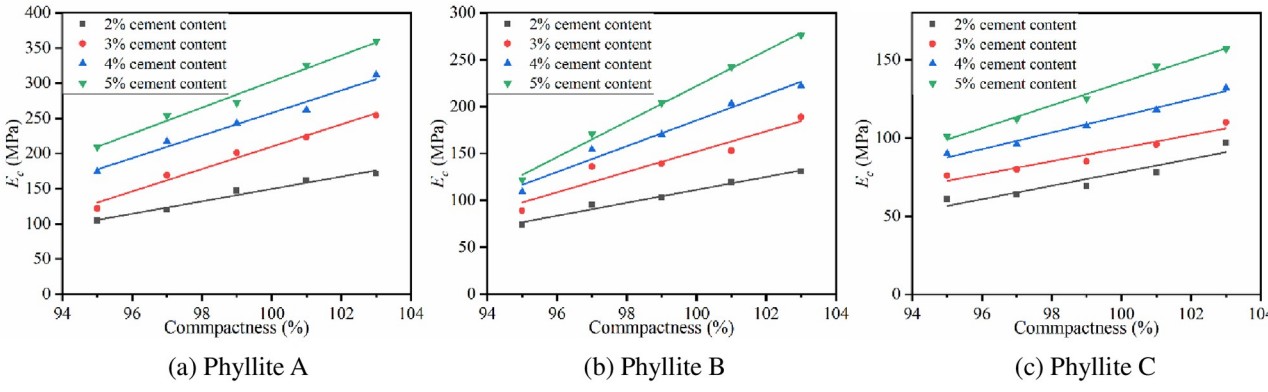

**Fig 11. Relationship between 7d $E_c$ and compactness.**

VVCM and static pressure method were used to prepare cement-improved phyllite specimens with different compactness. The cement content was 4%, and the specimen was cured for 7 days under standard curing conditions, and then, the $E_c$ values of the phyllite specimen were tested. The results are shown in Fig 12.

From the figure, it is clear that when the cement content is 4%, regardless of the compaction method, the 7d $E_c$ shows a linear growth trend with increase of compactness. The $E_c$ of the improved phyllite specimens formed by VVCM can be increased by 5% to 10% for every 1% increase in compactness, which is greater than the rise seen in the samples formed by the static pressure method. It shows that the compaction method also has a certain influence on the $E_c$ of the improved phyllite subgrade filler, and it also reflects the superiority of VVCM.

## 4. Field test analysis of improved phyllite filling subgrade

### 4.1 Test section

Field tests on the improvement of strongly weathered phyllite fillers were carried out in the K18+440—K18+540 and K20+220—K20+320 sections of the No. 4 bid section of the Ankang–Pingli expressway in Shaanxi. The test road sections were improved with cements of 2%, 3%, 4%, and 5%, and each sample was 50 m long.

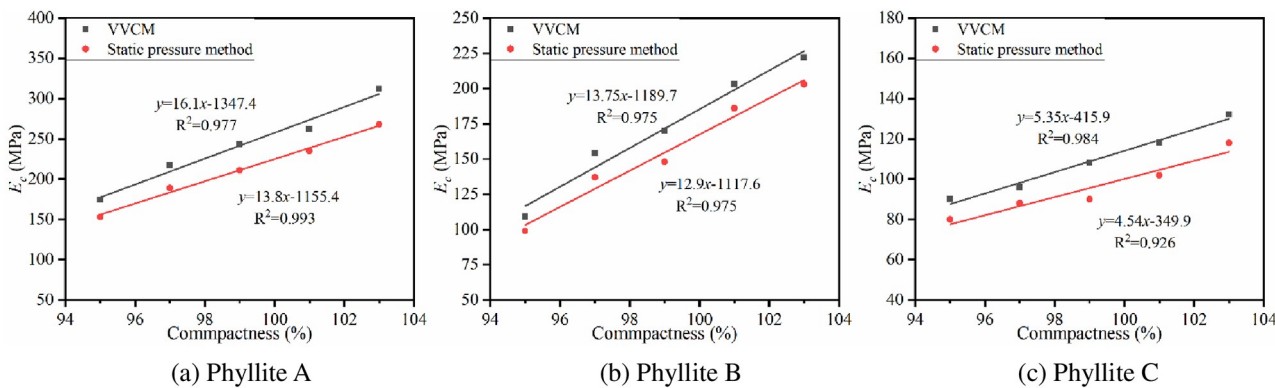

**Fig 12. The relationship between 7d $E_c$ and compaction method.**

**Table 9.** $E_c$ of strongly weathered phyllite filling subgrade under different cement content.

| Test bid | Cement content (%) | $E_c$ (MPa) |
|---|---|---|
| K18+440—K18+490 | 2 | 64.08 |
| K18+490—K18+540 | 3 | 79.15 |
| K20+220—K20+270 | 4 | 95.25 |
| K20+270—K20+320 | 5 | 108.47 |

## 4.2 On-site rebound deflection test method

$E_c$ is the main design parameter that reflects the strength, stiffness, and stability of subgrade, and it is also the main mechanical parameter that characterizes the ability of subgrade to resist deformation under traffic load. Therefore, according to the Field Test Methods of Subgrade Pavement for Highway Engineering (JTG 3450–2019) [40], the load-bearing plate method was adopted to test $E_c$ of the field subgrade.

## 4.3 Test result analysis

The sand-filling method was used to detect the compactness of the test section. The minimum compactness of the tested embankment cement-improved phyllite filler is 95.3%. It can meet the 94% compactness required by the Specifications for Design of Highway Subgrades (JTG D30-2015) [38]. $E_c$ tests were carried out on modified road sections with different cement content. The test results are summarized in Table 9.

The data in Table 9 show that $E_c$ of subgrade increases linearly with an increase in cement content. When the cement content is 2%, the subgrade $E_c$ is 64.08 MPa. When the cement content is 5%, the $E_c$ reaches 108.47 MPa, which is 69% higher than that of the subgrade with 2% cement. According to the Specifications for Design of Highway Asphalt Pavement (JTG D50-2017) [41], $E_c$ of highways and first-class highway subgrades should be 30 MPa. This shows that after the phyllite filler is improved by cement addition and that $E_c$ of the subgrade can meet the specification requirements. Further, comparison with the indoor test results shows that VVCM has better field correlation than the heavy compaction method, this is consistent with Yuan's research results [42].

## 5. Conclusions

1. The physical and mechanical properties of three typical phyllites along the Anping expressway were studied. The results show that as the degree of weathering increases, the porosity of the phyllite increases, and the effective contact area decreases, eventually leading to a continuous decrease in strength.

2. VVCM was proposed and used to evaluate the compaction characteristics of phyllite subgrade filling. The results show that the maximum dry density of the specimen formed by VVCM is larger and its optimal water content is lower.

3. The effects of compaction method and compactness on the mechanical properties of phyllite subgrade filling were studied. The results show that the $E_c$ and CBR of phyllite subgrade fillings increase with an increase in compactness. For every 1% increase in compactness, the $E_c$ of the phyllite specimen increased by at least 10%, and the CBR increased by at least 8%. In addition, $E_c$ and CBR of the VVCM specimen were greater than those of the specimen formed by the heavy compaction method. Thus, the merits of the VVCM were established.

4. The compaction and mechanical properties of cement-improved phyllite subgrade fillers were studied. The results show that with an increase in cement content, the maximum dry density and optimal water content both increases. In addition, the physical and mechanical properties of phyllite subgrade fillers improved greatly. The performance of cement-improved phyllite subgrade filler was also affected by the compactness and compaction method, and the performance of the VVCM-molded specimen was better than that of the specimen formed by the heavy compaction method.

5. Test subgrade $E_c$ was tested on the test section of the site, and the results showed that under identical loads, the larger the cement content, the smaller is the rebound deformation; the higher the subgrade strength. Using cement contents of 2%, 3%, 4%, and 5% improved phyllite subgrade specimens were obtained, and $E_c$ increased with the cement content.

## Supporting information

**S1 Data.**
(DOCX)

## Author Contributions

**Conceptualization:** Yingjun Jiang.

**Data curation:** Jiangtao Fan, Yong Yi, Kejia Yuan.

**Investigation:** Yingjun Jiang, Jiangtao Fan, Tian Tian.

**Methodology:** Jiangtao Fan, Yong Yi, Tian Tian, Changqing Deng.

**Supervision:** Kejia Yuan.

**Validation:** Yingjun Jiang, Jiangtao Fan, Yong Yi, Kejia Yuan.

**Visualization:** Yingjun Jiang, Tian Tian.

**Writing – original draft:** Yingjun Jiang, Jiangtao Fan.

**Writing – review & editing:** Changqing Deng.

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
