## [Decision Letter · Decision Letter 0]

19 Jan 2021

PONE-D-20-39942

Research on the mechanical properties of cement-improved phyllite based on the vertical vibration compaction method

PLOS ONE

Dear Dr. Fan,

Thank you for submitting your manuscript to PLOS ONE. After careful consideration, we feel that it has merit but does not fully meet PLOS ONE’s publication criteria as it currently stands. Therefore, we invite you to submit a revised version of the manuscript that addresses the points raised during the review process.

Please consider all the comments of all reviewers

Please make all required changes

Delete Figure1, Figure2, Figure3, and Figure5

We look forward to receiving your revised manuscript.

Kind regards,

Ahmed Mancy Mosa, Ph.D.

Academic Editor

PLOS ONE

Journal Requirements:

2. Please include the geographic coordinates of the site(s) from which Phyllite samples were collected.

Reviewers' comments:

Reviewer's Responses to Questions

**Comments to the Author**

1. Is the manuscript technically sound, and do the data support the conclusions?

Reviewer #1: Partly

Reviewer #2: Yes

Reviewer #3: Partly

Reviewer #4: No

Reviewer #5: Yes

2. Has the statistical analysis been performed appropriately and rigorously? 

Reviewer #1: N/A

Reviewer #2: Yes

Reviewer #3: N/A

Reviewer #4: No

Reviewer #5: N/A

3. Have the authors made all data underlying the findings in their manuscript fully available?

Reviewer #1: Yes

Reviewer #2: Yes

Reviewer #3: Yes

Reviewer #4: Yes

Reviewer #5: Yes

4. Is the manuscript presented in an intelligible fashion and written in standard English?

Reviewer #1: Yes

Reviewer #2: Yes

Reviewer #3: No

Reviewer #4: No

Reviewer #5: Yes

5. Review Comments to the Author

Reviewer #1: - More suitable title should be selected for the article. Title should decrease to 10-12 words.

- The abstract should state briefly the purpose of the research, the principal results and major conclusions. An abstract is often presented separately from the article, so it must be able to stand alone.

- It is suggested to present the structure of the article at the end of the introduction.

- The necessity and innovation of the article should be presented to the introduction.

- A flowchart should be added to the article to show the research methodology.

- It is suggested to add articles entitled “Arshid and Kamal. Appraisal of Bearing Capacity and Modulus of Subgrade Reaction of Refilled Soils” and “Fazelabdolabadi and Golestan. Towards Bayesian Quantification of Permeability in Micro-scale Porous Structures – The Database of Micro Networks” to the literature review.

- The major defect of this study is the debate or Argument is not clear stated in the introduction session. Hence, the contribution is weak in this manuscript. I would suggest the author to enhance your theoretical discussion and arrives your debate or argument.

- More suitable title should be selected for the figure 6 instead of “Comparison of maximum dry density between heavy compaction and VVCM tests”.

- It is suggested to compare the results of the present research with some similar studies which is done before.

- Following, you will find some new related references which should be added to literature review:

Alzaim et al. Effect of Modulus of Bituminous Layers and Utilization of Capping Layer on Weak Pavement Subgrades;

Majeed et al. Evaluation of Concrete with Partial Replacement of Cement by Waste Marble Powder.

- Page 3: the following paragraph is unclear, so please reorganize that:

“In the case of phyllite B and C, it was difficult to extract a complete rock sample on site owing to their high degree of weathering. Hence, only a point load strength test was performed for these samples. The point load test has no specific requirements on the shape of the specimen, and it does not require that the specimen must be a regular cylinder or cube.”

- Much more explanations and interpretations must be added for the Results, which are not enough.

- Please make sure your conclusions' section underscore the scientific value added of your paper, and/or the applicability of your findings/results, as indicated previously. Please revise your conclusion part into more details. Basically, you should enhance your contributions, limitations, underscore the scientific value added of your paper, and/or the applicability of your findings/results and future study in this session.

- “Notation” should be added to the article.

- DOI of the references must be added (you can use “" ext-link-type="uri" xlink:type="simple">https://crossref.org/").

Reviewer #2: The paper "Research on the mechanical properties of cement-improved phyllite based on the

vertical vibration compaction method " is very interesting and helpful for the reader. The paper is very well written and has readable structure. Therefore, it deserved to publish in the journal.

However, the following points need explanation.

1- Please provide the latest references in the literature review.

2- The text in the figures are not readable please increase the axis label size.

3- Please provide the mechanics of the crack patter.

4- Describe about the experimental case study in more detail.

Reviewer #3: The manuscript entitled "Research on the mechanical properties of cement-improved phyllite based on the vertical vibration compaction method" has been investigated in detail. The topic addressed in the manuscript is potentially interesting and the manuscript contains some practical meanings, however, there are some issues which should be addressed by the authors:

1) The readability and presentation of the study should be further improved. The paper suffers from language problems. The paper should be proofread by a native speaker or a proofreading agent.

2) The Introduction section needs a major revision in terms of providing more accurate and informative literature review and the pros and cons of the available approaches and how the proposed method is different comparatively. Also, the motivation and contribution should be stated more clearly.

3) The importance of the material carried out in this manuscript can be explained better than other important studies published in this field. I recommend the authors to review other recently developed works.

4) What makes the proposed material suitable for this unique mixture? What new findings to the proposed material have the authors added (compared to the existing experiments)? These points should be clarified.

5) "Conclusion" section should be edited in a more highlighting, argumentative way. The authors should analysis the reason why the tested results is achieved.

6) The main contributions of the study should be clearly explained in both theoretical and practical aspects.

7) The authors should clearly emphasize the contribution of the study. Please note that the up-to-date of references will contribute to the up-to-date of your manuscript.

8) It will be helpful to the readers if some discussions about insight of the main results are added as Remarks or Discussion.

This study may be proposed for publication if it is addressed in the specified problems.

Reviewer #4: The authors of this paper made an appreciable effort, however the manuscript is unable to elaborate the novelty and applicability of this research in the field. However, the quality of this manuscript can be improved by adding few more test, by discussing the findings and the applicability f the result on the site. Few comments are listed below:

1. The authors has mentioned that the Phyllite is very soft rock available on the study site and not suitable for subbase or subgrade, however, the addition of cement will improve the property. The increase in strength after adding cement is obvious but what is the durability of the stabilized materials? Durability test need to be added in the manuscript.

2. The Phyllite rock bed is available on the site but the test performed in the laboratory is based on the crushed rock. Authors does not discussed that how the cement will be mixed with the Phyllite rock on the site.

3. The title need to be rewritten. Do not start the title with the word "Research."

4. The abstract need to be rewritten and must include some part of Introduction, methodology, result and conclusion.

5. More citation in the introduction is required closely related to the present study.

Reviewer #5: 1. The title can be changed to be

Investigation on cement-improved phyllite based on the vertical vibration compaction method

2. Be more specific, where?? Line 9

3. The discerption of the material may be by grain size distribution not by long. Line78

4. The methodology and the laboratory work must be shortened to include only the important details. The number of designation of standards should be mentioned.

5. Figures 1-3 could be removed.

6. The conclusions covered the finding of the study.

7. The references are sufficient and recent.

8. The overall structure of the article is fair.

6. PLOS authors have the option to publish the peer review history of their article (what does this mean?). If published, this will include your full peer review and any attached files.

Reviewer #1: No

Reviewer #2: **Yes: **Afaq Ahmad

Reviewer #3: No

Reviewer #4: **Yes: **SHAMSHAD ALAM

Reviewer #5: No

---

## [Author Response · Author response to Decision Letter 0]

26 Jan 2021

Dear editor, 

I have revised the manuscript as required. Including the following:

1. Adjust the format of the manuscript to meet the requirements of PLOS ONE.

2. Deleted Figure 1, Figure 2, Figure 3 and Figure 5 in the manuscript.

3. The specific geographic coordinates of the phyllite sampling are along the Ankang-Pingli Expressway in Shaanxi Province, China.

4. A separate description is added for each figure in the manuscript.

The following is my specific response to the reviewers’ comments.

Dear reviewer 1,

Thank you again for reviewing and evaluating my manuscript, and the following is my reply and description.

Question 1: More suitable title should be selected for the article. Title should decrease to 10-12 words.

Response: Thanks for the reviewer’s comments. I have adjusted the title to 11 words. The current title reflects the core of this article.

Question 2: The abstract should state briefly the purpose of the research, the principal results and major conclusions. An abstract is often presented separately from the article, so it must be able to stand alone.

Response: Thanks for the reviewer’s comments. I have adjusted the abstract to make it an independent part. And delete the part except the purpose of the research, the principal results and major conclusions to make it simpler and clearer.

Question 3: It is suggested to present the structure of the article at the end of the introduction.

Response: Thanks for the reviewer’s comments. At the end of the introduction, I have summarized the research structure of the article to make it clearer.

Question 4: The necessity and innovation of the article should be presented to the introduction.

Response: Thanks for the reviewer’s comments. I have divided the last paragraph of the original introduction into two paragraphs. The first paragraph points out the current research status and deficiencies, and points out the necessity and innovation of this article. The second paragraph explains the research structure of the article.

Question 5: A flowchart should be added to the article to show the research methodology.

Response: Thanks for the reviewer’s comments. This article is a relatively conventional research paper. Except for VVCM, the test methods and compaction methods used are basically those widely used in the road industry. The research plan and the experimental methods used have been clearly presented in Section 2.3. So, I think that adding a flowchart is a bit repetitive, so I didn't add the flowchart to the manuscript.

Question 6: It is suggested to add articles entitled “Arshid and Kamal. Appraisal of Bearing Capacity and Modulus of Subgrade Reaction of Refilled Soils” and “Fazelabdolabadi and Golestan. Towards Bayesian Quantification of Permeability in Micro-scale Porous Structures – The Database of Micro Networks” to the literature review.

Response: Thanks for the reviewer’s comments. I have added "Arshid and Kamal. Appraisal of Bearing Capacity and Modulus of Subgrade Reaction of Refilled Soils" to the reference, but about "Fazelabdolabadi and Golestan. Towards Bayesian Quantification of Permeability in Micro-scale Porous Structures – The Database of Micro Networks", I did not find the source of the document. If it is convenient, can you provide me with the original text? I will add it to the reference after I read it.

Question 7: The major defect of this study is the debate or Argument is not clear stated in the introduction session. Hence, the contribution is weak in this manuscript. I would suggest the author to enhance your theoretical discussion and arrives your debate or argument.

Response: Thanks for the reviewer’s comments. I have readjusted the introduction debate to make it more convincing.

Question 8: More suitable title should be selected for the figure 6 instead of “Comparison of maximum dry density between heavy compaction and VVCM tests”.

Response: Thanks for the reviewer’s comments. The title of Figure 6 is indeed not appropriate, I have modified it to "Comparison of maximum dry density and optimal water content between heavy compaction and VVCM tests".

Question 9: It is suggested to compare the results of the present research with some similar studies which is done before.

Response: Thanks for the reviewer’s comments. I have added "Comparison of Mechanical Properties of Cement-Stabilized Loess Produced Using Different Compaction Methods" to prove that VVCM has a high correlation with the field, which is consistent with previous research results.

Question 10: Following, you will find some new related references which should be added to literature review: Alzaim et al. Effect of Modulus of Bituminous Layers and Utilization of Capping Layer on Weak Pavement Subgrades; Majeed et al. Evaluation of Concrete with Partial Replacement of Cement by Waste Marble Powder.

Response: Thanks for the reviewer’s comments. The two articles you provided are of great reference value, and I have added these two articles to the manuscript as references.

Question 11: Page 3: the following paragraph is unclear, so please reorganize that:“In the case of phyllite B and C, it was difficult to extract a complete rock sample on site owing to their high degree of weathering. Hence, only a point load strength test was performed for these samples. The point load test has no specific requirements on the shape of the specimen, and it does not require that the specimen must be a regular cylinder or cube.”

Response: Thanks for the reviewer’s comments. I have modified the third page of the manuscript to make it more fluent and easier to understand.

Question 12: Much more explanations and interpretations must be added for the Results, which are not enough.

Response: Thanks for the reviewer’s comments. What you said is of great value, and I have made the necessary adjustments in the manuscript.

Question 13: Please make sure your conclusions' section underscore the scientific value added of your paper, and/or the applicability of your findings/results, as indicated previously. Please revise your conclusion part into more details. Basically, you should enhance your contributions, limitations, underscore the scientific value added of your paper, and/or the applicability of your findings/results and future study in this session.

Response: Thanks for the reviewer’s comments. What you said is very important, and I have made the necessary adjustments in the manuscript. The purpose of this article is to illustrate the feasibility of phyllite as a subgrade filler, which is verified by a series of tests. Then in order to explain the superiority of the VVCM method compared with other methods, it is more relevant to the field core samples, which are explained in detail in the article and thesis.

Question 14: “Notation” should be added to the article.

Response: Thanks for the reviewer’s comments. I have made the necessary changes in the manuscript.

Question 15: DOI of the references must be added (you can use “https://crossref.org/").

Response: Thanks for the reviewer’s comments. I have added the DOI of all documents to the back of the corresponding references. For some conference papers without DOI, I have found the original link as a supplement.

Dear reviewer 2,

Thank you again for reviewing and evaluating my manuscript, and the following is my reply and description.

Question 1: Please provide the latest references in the literature review.

Response: Thanks for the reviewer’s comments. I have included some 2020 and 2021 articles in the references.

Question 2: The text in the figures is not readable please increase the axis label size.

Response: Thanks for the reviewer’s comments. The unreadable text of the picture is due to the stitching of the pictures, and formatting problems are likely to occur if they are placed separately. I have adjusted the picture in the manuscript to make it clearer.

Question 3: Please provide the mechanics of the crack patter.

Response: Thanks for the reviewer’s comments. The question you mentioned is very valuable. But I am very sorry that I did not understand exactly what you mean. If you are talking about the weathering cracks of phyllite A, B and C, then the principle is caused by the weathering of nature. If you are talking about cracks formed by the destruction of strength. Then the principle is that the upper and lower parts exert a force on the specimen. It produces shear deformation, resulting in damage. This is similar to the compressive strength failure in the reference "Engineering Comparison of Mechanical Properties of Cement-Stabilized Loess Produced Using Different Compaction Methods".

Question 4: Describe about the experimental case study in more detail.

Response: Thanks for the reviewer’s comments. I have made the necessary changes in the manuscript. One of the purposes of this article is to determine that VVCM is more relevant to the scene. Therefore, the field test research is based on this conclusion. Due to the limited conditions, there are not enough confirmatory field tests. I will explore this issue more in-depth in subsequent research.

Dear reviewer 3,

Thank you again for reviewing and evaluating my manuscript, and the following is my reply and description.

Question 1: The readability and presentation of the study should be further improved. The paper suffers from language problems. The paper should be proofread by a native speaker or a proofreading agent.

Response: Thanks for the reviewer’s comments. The question you raised is very valuable. In response to language issues, I have made necessary adjustments in the manuscript.

Question 2: The Introduction section needs a major revision in terms of providing more accurate and informative literature review and the pros and cons of the available approaches and how the proposed method is different comparatively. Also, the motivation and contribution should be stated more clearly.

Response: Thanks for the reviewer’s comments. I have adjusted the introduction in the manuscript. Added the latest references for explanation and demonstration. It also explains the shortcomings of the existing methods and the applicability of the proposed methods. I divided the original last paragraph into two paragraphs. The first part explains the shortcomings of the current method and the lack of relevance to the scene. The second part leads to my own research content and research ideas.

Question 3: The importance of the material carried out in this manuscript can be explained better than other important studies published in this field. I recommend the authors to review other recently developed works.

Response: Thanks for the reviewer’s comments. Your question is very valuable. I have added other latest literature on phyllite materials in the manuscript to better explain it.

Question 4: What makes the proposed material suitable for this unique mixture? What new findings to the proposed material have the authors added (compared to the existing experiments)? These points should be clarified.

Response: Thanks for the reviewer’s comments. Weathered phyllite is extremely difficult to meet the minimum requirements of subgrade filling specifications. But adding cement to improve can meet the requirements, which is also explained in the second part of the introduction. However, due to the low correlation between the heavy compaction method and the site, the results obtained from the indoor test pieces cannot be used to better guide the site practice. Most of the existing researches are based on the heavy compaction method, but the VVCM method can better establish a connection with the present, so as to judge whether the cement-improved phyllite can meet the requirements of on-site construction.

Question 5: "Conclusion" section should be edited in a more highlighting, argumentative way. The authors should analysis the reason why the tested results are achieved.

Response: Thanks for the reviewer’s comments. I have revised the conclusion in the manuscript. The reason for obtaining the test result has been explained and supplemented in the results and discussion.

Question 6: The main contributions of the study should be clearly explained in both theoretical and practical aspects.

Response: Thanks for the reviewer’s comments. I have made the necessary changes in the manuscript. For the theoretical explanation, I have added a few recent references to prove the correctness of the theory. Regarding practice, I am sorry that due to the limited site conditions, the collected data is not enough, so I quoted the previous research results to deepen the explanation of the site relevance. I will focus on this in future research.

Question 7: The authors should clearly emphasize the contribution of the study. Please note that the up-to-date of references will contribute to the up-to-date of your manuscript.

Response: Thanks for the reviewer’s comments. This question you mentioned is very valuable. I have added several recent research papers as references in the manuscript to enhance the value and contribution of this research.

Question 8: It will be helpful to the readers if some discussions about insight of the main results are added as Remarks or Discussion.

Response: Thanks for the reviewer’s comments. The question you mentioned is of great help to me, and I have made the necessary changes in the manuscript.

Dear reviewer 4,

Thank you again for reviewing and evaluating my manuscript, and the following is my reply and description.

Question 1: The authors has mentioned that the Phyllite is very soft rock available on the study site and not suitable for subbase or subgrade, however, the addition of cement will improve the property. The increase in strength after adding cement is obvious but what is the durability of the stabilized materials? Durability test need to be added in the manuscript.

Response: Thanks for the reviewer’s comments. What you said is very meaningful. I am sorry that I did not perform the durability test of cement modified phyllite in the article. This will be the focus of my follow-up research, and I will focus on this in subsequent research. In the introduction, I referred to previous research results, which indirectly demonstrated that cement-improved phyllite has better stability. And the improvement of a variety of materials by cement, such as loess, expansive soil and other soft rocks, has been shown to improve its performance.

Question 2: The Phyllite rock bed is available on the site but the test performed in the laboratory is based on the crushed rock. Authors does not discuss that how the cement will be mixed with the Phyllite rock on the site.

Response: Thanks for the reviewer’s comments. It can be seen from Figure 9 that phyllite with different weathering degrees can meet the specifications for different requirements for subgrade filling. The direct use of phyllite as subgrade filler on site also has higher requirements for compaction. During the indoor test, I first studied the phyllite without cement and judged whether it can meet the standard of subgrade filling, so the conclusion drawn in Figure 9 appeared. Then I studied the cement-improved phyllite, and judged the influence of different cement content, compaction degree, and molding method on it, so as to better guide the on-site construction. Therefore, indoor tests include both weathered phyllite and cement-modified phyllite.

For cement-improved phyllite, instead of directly using the phyllite materials along the line, the phyllite is further processed. In this process, cement is added for modification, and then transported back to the site for construction.

Question 3: The title needs to be rewritten. Do not start the title with the word "Research."

Response: Thanks for the reviewer’s comments. I have revised the title in the manuscript.

Question 4: The abstract need to be rewritten and must include some part of Introduction, methodology, result and conclusion.

Response: Thanks for the reviewer’s comments. I have rewritten the abstract, deleted some content not related to this research, and then highlighted the core and focus of the research.

Question 5: More citation in the introduction is required closely related to the present study.

Response: Thanks for the reviewer’s comments. The question you raised is very meaningful. I have added several recent studies as references in the manuscript, hoping to improve the research of this article.

Dear reviewer 5,

Thank you again for reviewing and evaluating my manuscript, and the following is my reply and description.

Question 1: The title can be changed to be“Investigation on cement-improved phyllite based on the vertical vibration compaction method”

Response: Thanks for the reviewer’s comments. I have revised the title in the manuscript.

Question 2: Be more specific, where?? Line 9

Response: Thanks for the reviewer’s comments. Sorry for not explaining clearly. Mountain highway here refers to the highway from Ankang to Pingli in Shaanxi Province. In fact, not only along this highway, but also along many highways in Northwest China, there will be a lot of weathered phyllite.

Question 3: The discerption of the material may be by grain size distribution not by long. Line78

Response: Thanks for the reviewer’s comments. For the classification of weathered phyllite, I did not use length to classify, but distinguished by its degree of weathering. The description of its length in the material part is to give readers a more intuitive feeling. Because the mechanical properties of phyllite with different weathering degrees are very different, this will also be involved in the following research. Because both strongly weathered phyllite and moderately weathered phyllite are extremely fragile, they are prone to fragmentation during sampling, so it is not accurate to judge by the particle size alone.

Question 4: The methodology and the laboratory work must be shortened to include only the important details. The number of designation of standards should be mentioned.

Response: Thanks for the reviewer’s comments. The question you raised is very valuable. I have revised the method part in the manuscript. However, in order to ensure that readers can clearly understand the experimental procedures, the various experiments performed must be fully presented in the manuscript, so the length is indeed relatively long, which is unavoidable. For the numbering of the standard, I have added it in the manuscript.

Question 5: Figures 1-3 could be removed.

Response: Thanks for the reviewer’s comments. Figure 1-3 is a bit redundant; I have deleted Figure 1-3 in the manuscript.

---

## [Decision Letter · Decision Letter 1]

5 Feb 2021

PONE-D-20-39942R1

Investigation on cement-improved phyllite based on the vertical vibration compaction method

PLOS ONE

Dear Dr. Fan,

Thank you for submitting your manuscript to PLOS ONE. After careful consideration, we feel that it has merit but does not fully meet PLOS ONE’s publication criteria as it currently stands. Therefore, we invite you to submit a revised version of the manuscript that addresses the points raised during the review process.

Please consider the comment of reviewer5 ( Apply unified numbering system for the sections and subsections)

We look forward to receiving your revised manuscript.

Kind regards,

Ahmed Mancy Mosa, Ph.D.

Academic Editor

PLOS ONE

Reviewers' comments:

Reviewer's Responses to Questions

**Comments to the Author**

1. If the authors have adequately addressed your comments raised in a previous round of review and you feel that this manuscript is now acceptable for publication, you may indicate that here to bypass the “Comments to the Author” section, enter your conflict of interest statement in the “Confidential to Editor” section, and submit your "Accept" recommendation.

Reviewer #1: All comments have been addressed

Reviewer #2: All comments have been addressed

Reviewer #3: All comments have been addressed

Reviewer #5: All comments have been addressed

2. Is the manuscript technically sound, and do the data support the conclusions?

Reviewer #1: Yes

Reviewer #2: Yes

Reviewer #3: Yes

Reviewer #5: Partly

3. Has the statistical analysis been performed appropriately and rigorously? 

Reviewer #1: Yes

Reviewer #2: Yes

Reviewer #3: Yes

Reviewer #5: N/A

4. Have the authors made all data underlying the findings in their manuscript fully available?

Reviewer #1: Yes

Reviewer #2: Yes

Reviewer #3: Yes

Reviewer #5: Yes

5. Is the manuscript presented in an intelligible fashion and written in standard English?

Reviewer #1: Yes

Reviewer #2: Yes

Reviewer #3: Yes

Reviewer #5: Yes

6. Review Comments to the Author

Reviewer #1: (No Response)

Reviewer #2: The authors responded well against the suggested comments, please accept this interesting manuscript.

Reviewer #3: The authors have responded correctly to all comments and the new version of manuscript has been significantly improved. I recommend it for publication in PLOS ONE.

Reviewer #5: Apply unified numbering system for the sections and subsections

All comments were considered

The authors of this paper made an appreciable effort

7. PLOS authors have the option to publish the peer review history of their article (what does this mean?). If published, this will include your full peer review and any attached files.

Reviewer #1: No

Reviewer #2: **Yes: **Afaq Ahmad

Reviewer #3: No

Reviewer #5: No

---

## [Author Response · Author response to Decision Letter 1]

5 Feb 2021

Dear editor,

First of all, thank you and all reviewers for your recognition of my article. I have numbered the sections and subsections uniformly in accordance with the requirements of reviewer 5.

Dear reviewer 5,

Thank you again for reviewing and evaluating my manuscript, and the following is my reply and description.

Question: Apply unified numbering system for the sections and subsections.

Response: Thanks for the reviewer’s comments. Sorry for this problem. In the first submission of the manuscript, I gave a unified number to the sections, but I saw that the article templates on the website did not use a unified number, so I deleted it in the second submission. This point you raised is very valuable. Without a uniform section number, the article will appear unorganized. I have added it to the manuscript.

---

## [Decision Letter · Decision Letter 2]

10 Feb 2021

Investigation on cement-improved phyllite based on the vertical vibration compaction method

PONE-D-20-39942R2

Dear Dr. Fan,

We’re pleased to inform you that your manuscript has been judged scientifically suitable for publication and will be formally accepted for publication once it meets all outstanding technical requirements.

Kind regards,

Ahmed Mancy Mosa, Ph.D.

Academic Editor

PLOS ONE

Additional Editor Comments (optional):

Reviewers' comments:

Reviewer's Responses to Questions

**Comments to the Author**

1. If the authors have adequately addressed your comments raised in a previous round of review and you feel that this manuscript is now acceptable for publication, you may indicate that here to bypass the “Comments to the Author” section, enter your conflict of interest statement in the “Confidential to Editor” section, and submit your "Accept" recommendation.

Reviewer #5: All comments have been addressed

2. Is the manuscript technically sound, and do the data support the conclusions?

Reviewer #5: Yes

3. Has the statistical analysis been performed appropriately and rigorously? 

Reviewer #5: N/A

4. Have the authors made all data underlying the findings in their manuscript fully available?

Reviewer #5: Yes

5. Is the manuscript presented in an intelligible fashion and written in standard English?

Reviewer #5: Yes

6. Review Comments to the Author

Reviewer #5: The authors of this paper made an appreciable effort

All comments have been addressed

Recommendation is accept

7. PLOS authors have the option to publish the peer review history of their article (what does this mean?). If published, this will include your full peer review and any attached files.

Reviewer #5: No

---

## [Editor Report · Acceptance letter]

19 Feb 2021

PONE-D-20-39942R2 

Investigation on cement-improved phyllite based on the vertical vibration compaction method 

Dear Dr. Fan:

I'm pleased to inform you that your manuscript has been deemed suitable for publication in PLOS ONE. Congratulations! Your manuscript is now with our production department. 

Kind regards, 

on behalf of

Dr. Ahmed Mancy Mosa 

Academic Editor

PLOS ONE